# Validity of the Actigraph-GT9X accelerometer for measuring steps and energy expenditures in heart failure patients

Jisu Kim[1], Jonathan Kenyon[1], Hayley Billingsley[1¤a], Natalie Bohmke[1¤b], Syed Imran Ahmed[1], Hannah Salmons[1], Jung-Min Lee[2,3], Danielle Kirkman[1], Salvatore Carbone[1], Youngdeok Kim[1]*

1 Department of Kinesiology and Health Science, Virginia Commonwealth University, Richmond, VA, United States of America, 2 Graduate School of Physical Education, Kyung Hee University (Global Campus), Yongin-si, Gyeonggi-do, Republic of Korea, 3 Sports Science Research Center, Kyung Hee University (Global Campus), Yongin-si, Gyeonggi-do, Republic of Korea

☯ These authors contributed equally to this work.
¤a Current address: Clinical Studies Engineer, Biotronik, Lake Oswego, OR, United States of America
¤b Current address: Department of Physiology and Aging, College of Medicine, University of Florida, Gainesville, FL, United States of America
* kimy13@vcu.edu

**Data Availability Statement:** The data used in the present study are provided in the Supporting Information files.

## Abstract

### Introduction

The ActiGraph (AG) accelerometer is widely used to assess physical activity (PA) in heart failure (HF) patients. However, the validity of the AG in this population remains unexplored.

### Objective

Therefore, this study examined the criterion validity of the AG-GT9X for measuring step counts (SC) and energy expenditure (EE) among HF patients.

### Methods

16 patients with HF with preserved ejection fraction (mean age = 60.3±12.1yrs) completed a total of 41 symptom-limited cardiopulmonary exercise tests on a treadmill across multiple time points (median (IQR) = 2.5 (1.5–3.5)). All participants wore the AG (model: GT9X) on both the right ankle and waist locations during the test. Manually counted steps and indirect calorimetry-derived EE served as criterion measures. AG-derived EE was estimated using six different prediction equations previously developed for waist-worn AG. AG-derived measurements were compared with criterion measurements by calculating correlation coefficients, equivalence tests with two one-sided tests, mean absolute percentage error (MAPE), percentage bias, and Bland-Altman plots using mixed models to account for the nested nature of repeated measures within subjects.

**Funding:** This study was partially supported by a Career Development Award (19CDA34660318 to Dr. Salvatore Carbone; and 19CDA34740002 to Dr. Danielle Kirkman) from the American Heart Association.

## Results

Ankle-worn AG-SC was significantly equivalent to the criterion ($p < .05$) and had lower MAPE (<10%) compared to the waist location, regardless of PA intensity level. Sasaki-EE was significantly equivalent to the criterion ($p < .05$), with the lowest percentage bias overall (0.7%).

## Conclusions

The ankle-worn AG-SC and Sasaki-EE showed better accuracy among HF patients in laboratory settings. Further research is warranted to cross-validate the results in different settings.

## Introduction

Heart Failure (HF) is a growing public health concern. The prevalence of HF has significantly increased in recent decades [1], and it is now estimated that approximately 50% of all HF cases are caused by diastolic dysfunction with a left ventricular ejection fraction (LVEF) $\geq$ 50%, which is referred to as HF with preserved ejection fraction (HFpEF) [2]. HFpEF is a major contributor to the rising trend in HF hospitalizations. In the United States, HF hospitalizations increased by 26% from 2013 to 2017, with 1.2 million hospitalizations among 924,000 HF patients [3]. This trend highlights the growing importance of HFpEF, which accounts for a substantial portion of HF cases. While there is still uncertainty about effective treatment strategies for HFpEF, there has been an increased effort to identify therapeutic strategies, including physical activity (PA) [4, 5], to reduce the symptoms and risks associated with HFpEF. PA is a modifiable lifestyle behavior that can reduce HF-related hospitalizations, enhance functional capacity, and improve health-related quality of life in HF patients [6–8]. However, research on PA measurement practices in HF patients is limited. This is a critical gap in the literature, as PA is a crucial non-pharmacologic treatment component for HF patients [9].

Accelerometers have been widely used to assess PA levels in clinical research settings for HFpEF patients [2]. The Actigraph (AG)-GT9X (ActiGraph Inc, Pensacola, FL, USA) is one of the commonly used accelerometer devices to assess physical activity measures. The AG-GT9X is a triaxial accelerometer that can measure acceleration in three individual orthogonal planes (i.e., vertical (VT), anteroposterior, and mediolateral) [10, 11]. It summarizes high temporal resolution triaxial acceleration data by calculating the vector magnitude (VM) to provide a comprehensive measure of human behavior and movement [12]. Acceleration data can be processed into activity counts, which can be used to obtain indicators to assess PA levels [13]. Step counts (SC) and estimated energy expenditure (EE) are commonly used indicators to quantify PA levels utilizing activity counts derived from acceleration data [10, 14–16]. SC is a simple and objective measure of activity counts with higher values indicating a greater amount of activity; while EE provides a more complex measure of PA intensity levels that reflects the amount of energy expended during PA.

Nevertheless, accurately assessing PA levels with AG is challenging due to several factors, including variations in SC across the placement of AG on the body (i.e., wrist, ankle, or waist), environment settings (i.e., laboratory or free-living), population demographics (i.e., younger or older adults), and the use of different prediction equations to estimate EE developed relying on waist-worn AG and targeting different population groups [14, 17, 18]. Variations in SC

measurements have commonly been observed across AG wear locations. It has been widely reported that the ankle and waist locations show better accuracy for detecting SC than the wrist location in both younger and older adults, particularly in laboratory settings [17, 19, 20]. Moreover, in previous research, the ankle-worn AG more accurately measured SC than the waist-worn AG [16, 21, 22]. However, one study found that the waist-worn AG was more accurate than the ankle-worn AG at higher walking speeds, while the ankle-worn AG was more accurate at lower walking speeds [20]. This suggested that the accuracy of the AG in measuring SC may vary depending on both the PA intensity (e.g., walking speed) and where the AG is worn anatomically. Although few studies have assessed the accuracy of the AG in the HFpEF population, these variances in SC measurement may also be present in this clinical population.

Additionally, various algorithms have been used to estimate EE using the AG, such as the Freedson, refined Crouter, Sasaki, or Santos-Lozano equations [11, 23–25]. These equations were developed with a regression model to predict EE using the AG acceleration count data and calibrated and validated on the locomotor activities compared with indirect calorimetry measurements [22, 26]. However, these equations were specifically developed using data obtained from waist-worn AG and mainly targeting young and middle-aged adults, which made their validity in predicting EE subject to debate depending on the PA types, intensity, or population groups [23, 24]. In clinical practice, the evidence for the validity of these EE prediction equations is often limited by differences in population characteristics, such as poor ambulatory patterns and severe obesity, which may differ compared to the healthy general population. For example, Mandigout et al. assessed the accuracy of the AG in estimating EE using the Crouter equation (2010) in post-stroke patients with poor walking ability (i.e., walking asymmetries), finding reduced accuracy in predicting EE within this population [27]. Similarly, Ribeiro et al. investigated the validity of three AG-EE prediction equations (i.e., Williams Work-Energy (1998), Freedson Combination (1998), and Freedson VM3 Combination (2011)) in severely obese women and found that using these AG prediction equations resulted in wide variation in estimated EE and poor agreement with doubly labelled water [28]. These insights emphasize the need to examine the validity of AG-EE prediction equations in HFpEF, considering the high prevalence of obesity and other cardiometabolic comorbidities within this population [29].

Furthermore, as most AG-EE prediction equations were designed for healthy adults, they may not be appropriate for the HFpEF population. This is because the exercise capacity of HFpEF patients is typically lower than that of healthy adults [30, 31], and the AG-EE prediction equations may not be able to predict EE in this population accurately. Therefore, the current study aimed to assess the criterion validity of the AG accelerometer in measuring SC and estimating EE using six different prediction equations during treadmill walking in patients with HFpEF. Specifically, this study compared the AG-derived SC with manually counted steps and evaluated the AG-derived EE with indirect calorimetry-measured EE. The results from the present study are expected to provide valuable implications for the clinical use of the AG in assessing PA levels in patients with HFpEF.

## Materials and methods

### Study participants

The current study was conducted as part of two clinical trials examining the effects of unsaturated fatty acids on cardiorespiratory fitness (UFA-Preserved2; NCT03966755) and the role of mitochondrial dysfunction on exercise capacity in patients with HFpEF (MitoP; NCT03960073). All participants provided written informed consent. Ethical approval was

provided by the Virginia Commonwealth University Institutional Review Board for both trials (UFA-Preserved2: HM20016253; MitoP: HM20015440). Both studies were carried out at the same institution during overlapping periods and utilized the same measurement protocols for cardiopulmonary exercise testing. The inclusion criteria for both trials were identical, except for the additional criterion of obesity in the UFA-Preserved2 trial. Sixteen adults aged 18 and over with a confirmed clinical diagnosis of stable HF (i.e., New York Heart Association (NYHA) class II–III) and LVEF > 50% documented in the prior 12 months were included in the current study. Of those, 15 participants came from UFA-Preserved2 trial, which has an additional criterion of Body Mass Index (BMI) $\geq$ 30 kg/m$^2$ or total body fat percentage >25% in men and >35% in women. The exclusion criteria differed slightly depending on the risks associated with experimental conditions for each trial. Overall, those who had absolute contra-indications to exercise testing according to the American College of Sports Medicine guide-lines [32], significant chronic diseases (i.e., angina, uncontrolled arterial hypertension, chronic pulmonary disease, current cancer, etc.), electrocardiography changes (i.e., ischemia or arrhythmias), comorbidity limiting survival, stage V kidney disease, unstable fluid overload, pregnancy, or inability to give informed consent were excluded.

## Procedures and measures

**Cardiopulmonary exercise testing (CPET).**   All participants underwent a supervised maximal effort, symptom-limited, incremental CPET on a treadmill. Study participants completed the CPET up to four times in the UFA trial and three times in the MitoP trial. Therefore, a total of 41 CPET data were collected from sixteen participants and included in this study. The median number of CPETs performed was 2.50, with an interquartile range of 1.5 to 3.5.

The test was conducted by a certified exercise physiologist, and breath by breath gas-analysis was performed with open-circuit spirometry using a metabolic cart (MGC Diagnostics; Ultima CardiO$_2$ gas exchange analysis system metabolic measurement system). The treadmill used a gradual ramping protocol beginning at a speed of 1.0 miles per hour (mph) and an incline of 0%, and every 30 seconds, the speed and incline were increased by 0.1 mph and 0.5%, respectively, leading to an approximate increase of 0.3 metabolic equivalents (METs) every 30 seconds [33, 34]. The test was performed until the participants reached volitional fatigue. Participants' oxygen consumption (VO$_2$, ml/kg/min) and METs were recorded for EE estimation during CPET. The METs values were calculated by dividing the VO$_2$ by the standard resting metabolic rate (3.5 ml/kg/min) [35].

After thoroughly reviewing each participant's raw dataset, the first and last 5% of the data were eliminated before analysis and retained a steady state during CPET to minimize the variability from the potential uncontrollable sources [14, 36].

**Manually-counted steps.**   One investigator (JK) manually counted the steps during CPET using the ALLCounter android app (version 1.2.3; digicraft.org). The time-stamped manually counted steps data were downloaded as CSV files and matched with AG-derived SC data. The rater agreement was checked with an additional rater (YK) during the first three CPET tests, and there was greater than 99.5% agreement between the raters.

**ActiGraph GT9X accelerometer.**   Each participant wore AG-GT9X accelerometers (firm-ware v1.7.1, ActiGraph Inc, Pensacola, FL, USA) strapped to the right hip and ankle with an elastic band during CPET. The GT9X is one of the models of AG accelerometers, capable of measuring acceleration counts in three axes (X, Y (vertical on waist), and Z) within a dynamic range of ± 8 g, operating at sampling frequencies ranging from 30 to 100 Hz [12, 37]. For the present study, the GT9X was initialized with a 60 Hz sampling frequency with the optional Idle Sleep Mode feature off. Accelerometer data were downloaded in raw format as.gt3x files

and then converted into 1-, 10-, and 60-second epochs data using the normal filter in ActiLife software v. 6.13.3 (ActiGraph Inc, Pensacola, FL, USA) to remove high-frequency noise while preserving the relevant signal. The data was saved as CSV files to match the CPET data with the start and end times of the exercise test.

Step counts obtained from the ankle- and waist-worn AG were compared to manually counted steps. Additionally, EEs estimated from the six different algorithms used with the waist-worn AG were compared to indirect calorimetry-derived EE during CPET. To assess the validity of the AG for estimating EE, we used six different EE prediction equations (Freedson, Freedson Combination, Sasaki, Santos-Lozano VT, and Santos Lozano VM) developed using waist-worn AG accelerometers [11, 23–25]. The Freedson equation (1998), the most widely used in large-scale studies, was developed to predict EE using 60-second epochs for the VT axis based on a laboratory setting with a treadmill exercise [23]. The Freedson Combination equation (1998) combines the William Work-Energy equation (1998) to estimate EE below 1951 counts and the Freedson equation (1998) to estimate EE above 1952 counts using 60-second epochs [12, 28]. The refined Crouter equation (2010) is a two-regression model that uses coefficient variation, examining the variability among six consecutive 10-second epochs to differentiate resting, walking, jogging, and lifestyle activities during a minute [24]. As the study used a laboratory setting with CPET, the adapted version of the refined Crouter equation, specifically developed for walking/running (i.e., if the VT counts·10-second$^{-1}$ >8, and the coefficient variation of the counts per 10 seconds is ≤10) was used in the current study [12, 14]. The Sasaki equation (2011), which uses VM activity count, was developed to predict EE using 60-second epochs for adults in laboratory settings with a treadmill [11]. Lastly, the Santos-Lozano equations (2013) were developed to predict EE using 60-second epochs in laboratory settings for specific age groups (i.e., children, adults, and older adults) using the VT axis (Santos-Lozano VT) and the VM (Santos-Lozano VM) [25]. Since only the refined Crouter equation used 10-second epochs, we also presented Crouter-EE predicted in 60 seconds by summarizing 10-second epochs data into 60 seconds to allow for comparison with other equations in our study.

## Statistical analysis

We first conducted scatter plots to examine the linear relationship between AG-derived and criterion-derived measurements. Total step counts and mean METs were calculated and compared between the measurements (i.e., AG vs. criterion) using a mixed model, with a random intercept accounting for multiple observations within each participant. The correlation coefficients between the AG-derived and criterion-derived measurements were also estimated using a mixed model with a random intercept, according to Hanlett et al [38]. We used the bootstrapping method to calculate the bias-corrected 95% confidence intervals, with 200 bootstrap resamples generated from the observations at the individual level. We then calculated a mean CPET duration (minutes), VO2 ml/kg/min, total steps (counts), and EE estimations (METs) across different PA intensity levels and summarized them for comparison between AG-derived and criterion-derived measurements. PA intensity levels were categorized into light-intensity PA (LPA; 1.50–2.99 METs) and moderate-to-vigorous intensity PA (MVPA; ≥3 METs) based on PA guidelines for Americans [39], using EE estimation data obtained from indirect calorimetry during CPET.

We also performed the equivalences test using the two one-sided t-tests (TOST) to determine if AG-derived SC and EE estimations were significantly equivalent to the criterion measures. Specifically, we compared the mean ratio of SC and EE estimates between AG and criterion measures with the upper and lower limits of 10% equivalence zones (i.e., Ha$_1$: 0.9<

mean ratio and Ha$_2$: mean ratio<1.11) [37, 40]. The manually counted steps and indirect calorimetry-derived EE were used as criterion measures when creating 10% equivalence zones. Considering the multiple observations within each individual, a mixed model with a random intercept was used to test the mean of AG-derived measures against zero. We determined that PA estimates from AG were significantly equivalent to the criterion measures if the one-sided $p$-values from both tests were less than 0.05 (i.e., two-sided $p$-value estimation was divided by 2 to obtain the one-sided $p$-value when the hypothesized direction has met) [37, 40].

The accuracy of AG-derived SC and EE estimates compared to criterion measures was examined using mean absolute percentage error (MAPE; %), percentage bias (%), mean bias, and the Bland-Altman method. We have determined that MAPE <10% is an acceptable level of accuracy in the current study based on the accuracy standard of wearable PA monitors established by the Consumer Technology Association [41]. In addition, we conducted the modified Bland-Altman plots with a mean bias and 95% limit of agreement (LOA) to assess the agreement between AG-derived and criterion-derived measurements. The mean bias and 95% LOA were calculated using a one-way random effects model, taking into account repeated measures nested within a random factor of the subject. LOA is defined by the true difference between measurements (i.e., AG–criterion) accounting for random variability between- and within-subjects. Additionally, we calculated the 95% confidence intervals of LOA using the method of variance estimate recovery (MOVER), which provided estimates for the maximum limits of lower and upper limits of LOA. The MOVER approach is considered more effective than the conventional delta method with the data involving multiple observations per individual [42]. A detailed description with a SAS macro employing the Bland-Altman method using repeated measures within subjects is followed by Zou [42]. The modified Bland-Altman plots differ from the original method by using the criterion measures on the X-axis against the differences between the two measurements on the Y-axis, which allows for examining the accuracy of AG compared to the criterion measures [43]. The proportional bias was also assessed by examining the linear association between AG-derived and criterion-derived measurements in modified Bland-Altman plots. All statistical analyses were performed using SAS v9.4 (SAS Institute, Cary, NC, USA), and the statistical significance was set at $p<0.05$.

## Results

A detailed description of all test measurements used in the current study is provided in Table 1.

Sixteen adults (male = 1; female = 15; mean age = 60.25 ± 12.09 years) were included in this study. The participants included both Whites and Blacks, with the majority classified as NYHA class II, which is a classification of HF determined by the New York Heart Association, class II indicating a patient has mild symptoms and slight limitations on PA [44]. The mean LVEF (%) was 59.41± 3.58%, and the mean BMI was 38.74 ± 7.04 kg/m$^2$. Further demographic information for the entire sample is provided in Table 2, and the glossary of abbreviations is described in the S1 Table.

The scatter plots with correlation coefficient estimations between AG-derived and criterion measures-derived PA estimates (i.e., SC and EE) are presented in Figs 1 and 2. The positive relationships and strong correlations were observed between all AG-derived total SCs and manually counted total steps (Ankle: $r$ = 0.948, 95%CI = 0.943–0.952; Waist: $r$ = 0.923, 95% CI = 0.918–0.928; Fig 1). In addition to EE, moderate to strong correlations were observed between all AG-derived mean EEs and indirect calorimetry-derived mean EEs ($r$'s = 0.500–0.701; Fig 2).

**Table 1. Summary of all test measurements included in the study.**

| Test measurements | Descriptions |
|---|---|
| **Cardiopulmonary Exercise Testing (CPET)** | |
| Steps (counts) | Manually counted step (ALLcounter hand-tally app) |
| Energy expenditure (METs) | Indirect calorimetry |
| **ActiGraph GT9X** | |
| Steps (counts) | Ankle/Waist locations |
| Energy Expenditure (METs) | Waist location |
| **AG-EE prediction equations (cut-points; METs)** | |
| Freedson | $1.439008+0.000795\times$(VT cpm) |
| Freedson Combination | $1.439008+0.000795\times$(VT cpm)$\times$(cpm$>$1951) |
| | $0.0000191\times$(VT cpm)$\times$(weight kg)$\times$(cpm$\leq$1951) |
| Refined Crouter [a, b] | $2.294275\times$(exp $(0.00084679\times$VT cnts$\times10$sec$^{-1}$)) |
| Sasaki | $0.000863\times$(VM cpm)$+0.668876$ |
| Santos-Lozano VT | $2.8867+0.00067\times$(VT cpm)$-0.6807\times$Gender |
| Santos-Lozano VM | $2.5878+0.00047\times$(VM cpm)$-0.6453\times$Gender |

EE = energy expenditure; METs = Metabolic equivalents; AG = ActiGraph; VT = Vertical axis; cpm = count per minute; cnts = counts; CV = Coefficient of Variation of the vertical axis in 10 seconds; VM = Vector Magnitude.
[a]The lowest coefficient of variation (CV) is selected from those calculated for each 10-sec epoch, and all combinations of the five surrounding 10-sec epochs.
[b]All equations except for the Refined Crouter equation used 60-sec epoch activity count data for EE estimation. The Refined Crouter equation used 10-sec epoch activity count data.

Mean CPET duration (minutes), VO$_2$ ml/kg/min, and PA estimations according to different PA intensity levels (i.e., light and moderate-to-vigorous) were calculated and compared between AG and criterion measures using the equivalences test (Table 3). According to the equivalences test, SC from the ankle-worn AG was found to be significantly equivalent to manually counted steps both overall and across all levels of PA intensity ($p$'s$<0.05$; Table 3). SC from the waist-worn AG did not yield significant equivalence to manually counted steps, regardless of PA intensity levels.

For overall EE estimation, the refined Crouter, Sasaki, and Santos-Lozano VT/VM equations-EE were significantly equivalent to the indirect calorimetry-EE ($p$'s$<0.05$; Table 3). The Freedson, refined Crouter (10 sec/60 sec), and Sasaki equations-EE were significantly equivalent to the indirect calorimetry-EE at LPA ($p$'s$<0.05$; Table 3), while the Freedson Combination, Sasaki, and Santos-Lozano VT/VM equations-EE were significantly equivalent to indirect calorimetry-EE at MVPA ($p$'s$<0.05$; Table 3). Only the Sasaki equation-EE was significantly equivalent to the indirect calorimetry-EE, regardless of PA intensity.

The accuracy of AG-derived PA estimates compared to the criterion measures was examined using MAPE, percentage bias, and mean bias with 95% LOA in Table 4. Overall, SC from ankle-worn AG had lower MAPE [Ankle vs. Waist (%); Overall: 8.51 vs. 30.09; LPA: 10.72 vs. 43.71; MVPA: 6.28 vs. 16.75], lower percentage bias [Ankle vs. Waist (%); Overall: -6.16 vs. -29.14; LPA: -8.16 vs. -43.21; MVPA: -4.14 vs. -15.36], and lower mean bias [Ankle vs. Waist; Overall: -0.98 vs. -4.27; LPA: -1.36 vs. -6.13; MVPA: -0.74 vs. -2.67] than the waist-worn AG, regardless of PA intensity level. For EE estimation, the refined Crouter (60sec)-EE had the lowest MAPE overall and LPA [MAPE (%); Overall: 13.18; LPA: 11.81], while Santos-Lozano VT-EE had the lowest MAPE at MVPA [MAPE (%): 9.41], followed by Santos-Lozano VM-EE [MAPE (%): 9.99]. Unlike, the Sasaki-EE showed the lowest bias overall and at LPA [Overall: percentage bias (%): 0.65; mean bias: 0.02; LPA: percentage bias (%): -1.19; mean bias: -0.07],

**Table 2. Descriptive characteristics of the study participants.**

| | Total sample (N = 16) |
|---|---|
| Repeat tests (times) | |
| Total # | 41 |
| Median (IQR) | 2.5 (1.5–3.5) |
| Age (years) | 60.25 ± 12.09 |
| Gender (n, %) | |
| Male | 1 (6.25%) |
| Female | 15 (93.75%) |
| Race/Ethnicity (n, %) | |
| White | 8 (50.00%) |
| Black | 8 (50.00%) |
| NYHA classification (n, %) | |
| Class II | 13 (81.25%) |
| Class III | 3 (18.75%) |
| LVEF (%) | 59.41 ± 3.58 |
| Height (cm) | 164.80 ± 8.00 |
| Weight (kg) | 106.02 ± 25.11 |
| BMI (kg/m$^2$) | 38.74 ± 7.04 |
| Blood pressure | |
| SBP (mm Hg) | 128.00 ± 11.98 |
| DBP (mm Hg) | 69.38 ± 7.47 |
| Resting HR (beats/min) | 75.88 ± 10.11 |
| Major comorbidities (n, %) | |
| Diabetes | 9 (56.25%) |
| Ischemic heart disease | 1 (6.25%) |
| Biomarkers | |
| NT-proBNP (pg/ml), median (IQR) | 71 (50–249) |

BMI = body mass index; NYHA = New York Heart Association classification of the extent of heart failure;

LVEF = Left Ventricular Ejection Fraction. IQR = Interquartile Range; SBP = Systolic Blood Pressure;

DBP = Diastolic Blood Pressure; HR = Heart Rate; NT-proBNP = N-terminal pro-brain natriuretic.

Values are presented as mean ± standard deviations if normally distributed or median (IQR) if normality assumption is violated unless otherwise specified.

while the Santos-Lozano VT equation showed the lowest bias at MVPA [percentage bias (%): -0.16; mean bias: -0.02].

Bland-Altman plots were generated with the 95% LOA and maximum limits of LOA to assess the agreement between AG and criterion measures in Figs 3 and 4. Significant proportional bias was observed for both the ankle- and waist-worn AG-derived SC [Ankle: $b$ (SE) = -0.06 (0.02), $p$ = .006; Waist: $b$ (SE) = 0.50 (0.03); $p < .001$; Fig 3]. In addition, significant proportional bias was observed for all AG-derived EEs ($p$'s<0.001) except for the Freedson Combination-EE [$b$ (SE) = 0.01 (0.05); $p$ = 0.878; Fig 4]. The results obtained from the Bland-Altman analysis for all PA estimates were summarized in the S2 Table.

## Discussion

Despite evidence that wearable device-measured PA is significantly associated with HF patients' health outcomes [45], there is insufficient evidence on the validity of AG accelerometers for use among the clinical population, including HFpEF patients. Therefore, our study

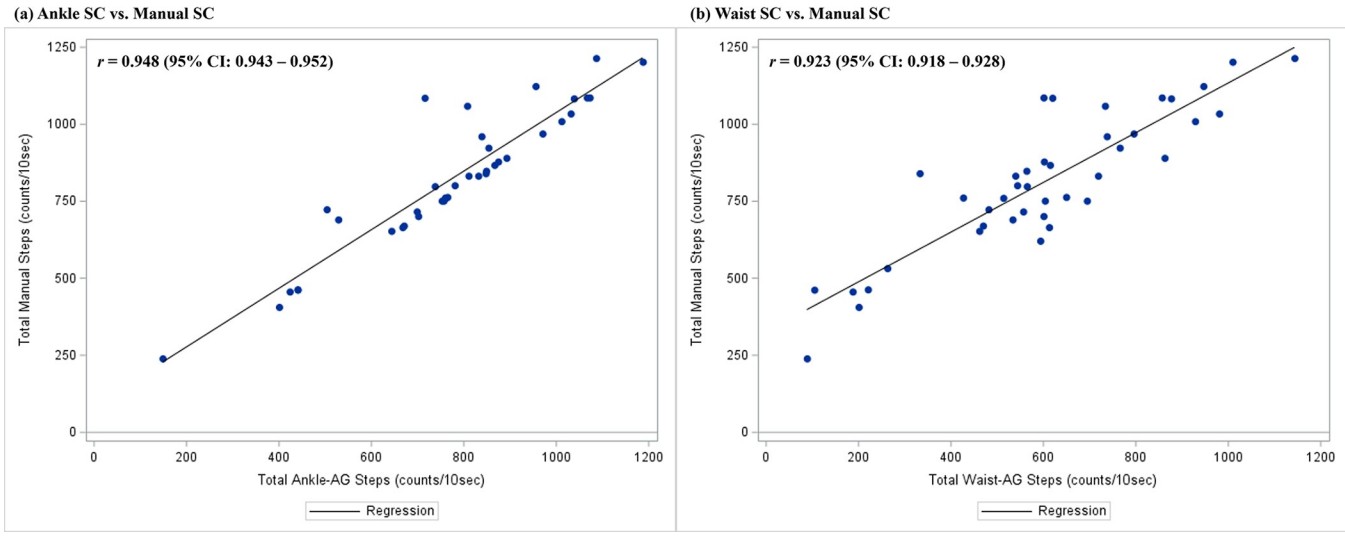

**Fig 1. Correlations between ActiGraph-derived total step counts (per 10-sec) and manual total step counts (per 10-sec).** (a) Ankle location; (b) Waist location. The diagonal line indicates a linear regression line.

aimed to evaluate the validity of the AG GT9X accelerometer for measuring SC and estimating EE during treadmill exercise testing at varying levels of PA intensity among individuals with HFpEF.

## Step counts

The main finding of this study was that the ankle-worn AG was more accurate in measuring SC in HFpEF patients than the waist-worn AG. The total SC obtained from the ankle-worn

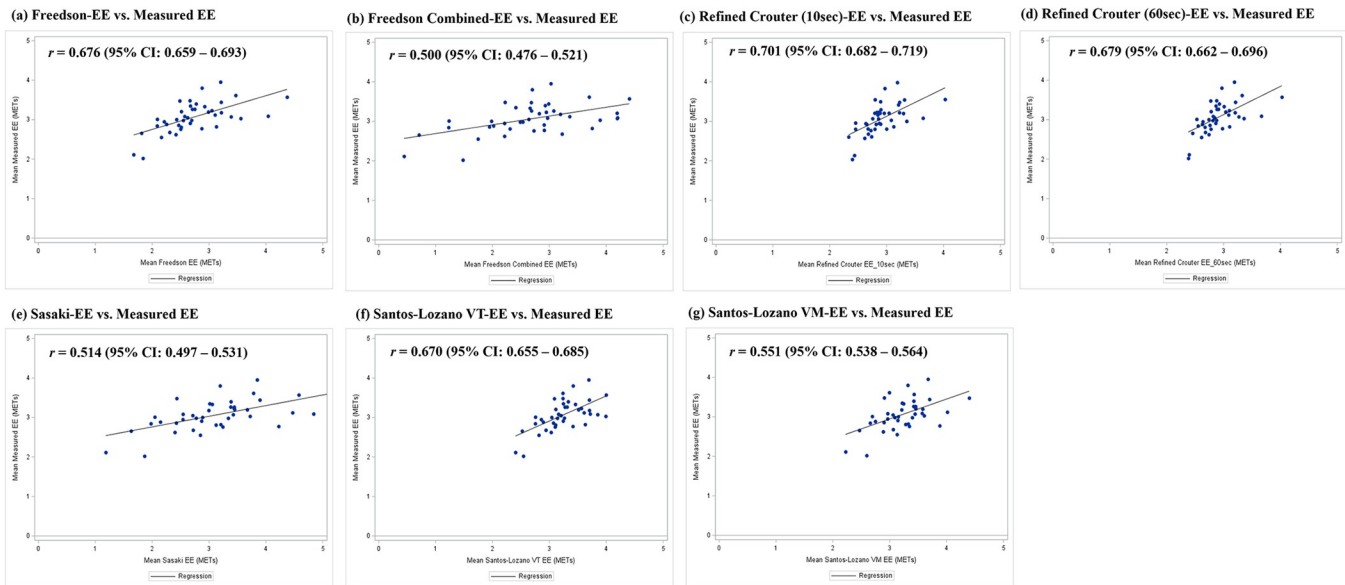

**Fig 2. Correlations between ActiGraph-derived mean energy expenditure (METs) using six different equations and indirect calorimetry-derived mean energy expenditure (METs).** (a) Freedson; (b) Freedson+Williams; (c) Refined Crouter using 10-sec epoch data; (d) Refined Crouter using 60-sec epoch data; (e) Sasaki; (f) Santos-Lozano VT; (g) Santos-Lozano VM. The Refined Crouter equation used 10-sec and 60-sec epoch data, while other equations used 60-sec epoch data. The diagonal line indicates a linear regression line.

**Table 3. Total step counts and energy expenditure during cardiopulmonary treadmill test.**

| | Overall[a] | LPA[a] | MVPA[a] |
|---|---|---|---|
| **Criterion measure** | | | |
| CPET duration (minutes) | 8.63 (7.52–9.74) | 4.16 (3.81–4.52) | 4.67 (3.64–5.69) |
| VO$_2$ ml/kg/min | 10.79 (10.12–11.46) | 8.38 (8.20–8.56) | 12.99 (12.35–13.64) |
| METs | 3.08 (2.89–3.27) | 2.41 (2.31–2.51) | 3.71 (3.53–3.90) |
| Manual step counts | 808.99(689.36–928.61) | 347.86 (312.83–382.88) | 478.81(336.76–590.87) |
| **ActiGraph GT9X** | | | |
| **Total Steps (counts)** | | | |
| Ankle-worn | 770.62 (665.24–875.99)* | 323.36 (279.65–367.08)* | 478.23 (379.65–576.81)* |
| Waist-worn | 603.07 (475.74–730.41) | 198.12 (173.60–222.65) | 418.00 (308.61–527.38) |
| **Energy expenditure (METs)[c]** | | | |
| Freedson | 2.76 (2.50–3.02) | 2.24 (2.09–2.40)* | 3.22(2.92–3.52) |
| Freedson Combination | 2.71 (2.32–3.09) | 1.99 (1.54–2.43) | 3.37 (3.04–3.69)* |
| Refined Crouter (10sec) [b] | 2.92 (2.76–3.07)* | 2.62 (2.49–2.74)* | 3.18 (2.99–3.30) |
| Refined Crouter (60sec) [b] | 2.93 (2.77–3.08)* | 2.64 (2.56–2.72)* | 3.18 (2.99–3.37) |
| Sasaki | 3.15 (2.80–3.51)* | 2.46 (2.20–2.72)* | 3.77 (3.35–4.19)* |
| Santos-Lozano VT | 3.28 (3.11–3.44)* | 2.84 (2.72–2.95) | 3.67 (3.48–3.86)* |
| Santos-Lozano VM | 3.25 (3.09–3.42)* | 2.87 (2.72–3.03) | 3.59 (3.40–3.78)* |

LPA = Light-intensity Physical Activity (1.50–2.99 METs); MVPA = Moderate-to-Vigorous Physical Activity ($\geq$3 METs); VT = Vertical Axis; VM = Vector Magnitude.

[a]Values are presented as mean (95% confidence intervals) from a random effect mixed model.

[b]The Refined Crouter equation used both 10 sec- and 60 sec-epoch data to estimate energy expenditure.

[c]AG-derived EEs were predicted from waist-worn AG accelerometers.

* Significantly equivalent with the criterion measure at 10% equivalence zone based on two one-sided *t*-tests equivalence test ($p < 0.05$).

AG was significantly similar to manually counted steps and showed lower values of MAPE (<10%), percentage bias, and mean bias with narrow 95% LOA than the waist-worn AG, regardless of PA intensity level. These findings are consistent with previous studies, which have demonstrated that the ankle location is the most accurate for SC detection as compared to the waist location [16, 22, 46]. It is generally considered that the accuracy of wearable devices should perform within MAPE values of less than 10%, indicating a high accuracy [41, 47]. For instance, in adults, Mora-Gonzalez et al. reported a lower MAPE for the ankle location than the waist location overall during the treadmill test (Ankle: 3% vs. Waist: 28%) [46]. Similarly, the ankle-worn AG in our study showed a lower MAPE than the waist-worn AG overall (Ankle: 8.5% vs. Waist: 30.1%). Additionally, in older adults, Korpan et al. found that the ankle location had a lower MAPE than the waist location (Ankle: 2.1% vs. Waist: 23.1%) [16], and Webber and St. John also reported that ankle location had a lower median absolute percentage error than waist location in older geriatric rehabilitation patients (Ankle: 2.5% vs. Waist: 18.9%) [22]. These findings are comparable to those of our study, as the mean age of our study population was over 60 years.

However, some previous studies have reported inconsistent results based on varying walking speeds. For instance, Karaca et al. found that the waist-worn AG was more accurate and had better agreement at higher walking speeds (6, 8, and 10 km/hr; MAPE (%): 1.2, 0.8, and 2.9, respectively) as compared to the ankle-worn AG (6, 8, and 10 km/hr; MAPE (%): 4.9, 47.7, and 50.6, respectively) [20]. In contrast, the ankle-worn AG was more accurate and showed better agreement at lower walking speeds (2 and 4 km/hr; MAPE (%): 12.4 and 1.0%) than the

**Table 4. Accuracy and agreement of AG-derived step counts and energy expenditures compared to criterion measures.**

| | MAPE (%) [a] | % bias [a] | Mean bias | 95% LOA | |
|---|---|---|---|---|---|
| | | | | Lower | Upper |
| **Step counts** | | | | | |
| **Ankle SC** | | | | | |
| Overall | 8.51 (5.12–11.90) | -6.16 (-9.83– -2.50) | -0.98 | -5.80 | 3.83 |
| LPA | 10.72 (7.40–14.03) | -8.16 (-11.73– -4.59) | -1.36 | -6.85 | 4.14 |
| MVPA | 6.28 (2.97–9.60) | -4.14 (-7.71– -0.57) | -0.74 | -5.80 | 4.33 |
| **Waist SC** | | | | | |
| Overall | 30.09 (22.54–37.63) | -29.14 (-37.11– -21.17) | -4.27 | -12.54 | 4.00 |
| LPA | 43.71 (37.64–49.77) | -43.21 (-49.66– -36.75) | -6.13 | -14.45 | 2.18 |
| MVPA | 16.75 (10.68–22.81) | -15.36 (-21.81– -8.91) | -2.67 | -9.83 | 4.49 |
| **Energy expenditure** | | | | | |
| **Freedson** | | | | | |
| Overall | 16.00 (13.51–18.49) | -10.84 (-16.53– -5.16) | -0.36 | -1.40 | 0.69 |
| LPA | 15.65 (13.11–18.19) | -9.00 (-14.48– -3.52) | -0.27 | -1.10 | 0.57 |
| MVPA | 16.34 (13.81–18.87) | -12.66 (-18.14– -7.19) | -0.47 | -1.60 | 0.66 |
| **Freedson Combination** | | | | | |
| Overall | 27.66 (21.59–33.74) | -14.43 (-26.73– -2.14) | -0.42 | -2.17 | 1.33 |
| LPA | 40.22 (34.29–46.15) | -18.48 (-30.26– -6.70) | -0.55 | -2.70 | 1.61 |
| MVPA | 15.28 (9.38–21.19) | -10.47 (-22.24–1.30) | -0.29 | -1.64 | 1.05 |
| **Refined Crouter (10sec) [b]** | | | | | |
| Overall | 16.08 (14.77–17.39) | -1.37 (-5.53–2.79) | -0.18 | -1.39 | 1.03 |
| LPA | 16.83 (15.46–18.20) | 11.54 (8.10–14.97) | 0.22 | -0.55 | 0.99 |
| MVPA | 15.35 (14.00–16.71) | -14.00 (-17.43– -10.57) | -0.55 | -1.58 | 0.48 |
| **Refined Crouter (60sec) [b]** | | | | | |
| Overall | 13.18 (11.72–14.64) | -2.87 (-6.96–1.22) | -0.19 | -1.23 | 0.86 |
| LPA | 11.81 (10.09–13.53) | 7.36 (3.77–10.94) | 0.14 | -0.48 | 0.76 |
| MVPA | 14.49 (12.82–16.17) | -12.92 (-16.49– -9.34) | -0.50 | -1.42 | 0.42 |
| **Sasaki** | | | | | |
| Overall | 17.92 (13.04–22.81) | 0.65 (-9.08–10.37) | 0.02 | -1.50 | 1.54 |
| LPA | 19.19 (14.32–24.07) | -1.19 (-10.42–8.05) | -0.07 | -1.47 | 1.34 |
| MVPA | 16.67 (11.81–21.53) | 2.45 (-6.78–11.67) | 0.07 | -1.51 | 1.65 |
| **Santos-Lozano VT** | | | | | |
| Overall | 13.06 (10.23–15.88) | 7.67 (3.66–11.68) | 0.16 | -0.74 | 1.06 |
| LPA | 16.80 (14.03–19.56) | 15.36 (11.78–18.95) | 0.33 | -0.34 | 1.01 |
| MVPA | 9.41 (6.66–12.16) | 0.16 (-3.41–3.73) | -0.02 | -0.92 | 0.87 |
| **Santos-Lozano VM** | | | | | |
| Overall | 14.73 (11.62–17.84) | 7.47 (1.31–13.63) | 0.14 | -0.87 | 1.14 |
| LPA | 19.60 (16.56–22.64) | 16.43 (11.22–21.64) | 0.37 | -0.49 | 1.22 |
| MVPA | 9.99 (6.97–13.01) | -1.34 (-6.54–3.86) | -0.09 | -0.98 | 0.81 |

AG = ActiGraph; SC = Step Counts; MAPE = Mean Absolute Percentage Error; LOA = Limit of Agreement; LPA = Light-intensity Physical Activity (1.50–2.99 METs); MVPA = Moderate-to-Vigorous Physical Activity ($\geq$3 METs); VT = Vertical axis; VM = Vector Magnitude.

[a] Values are presented as mean (95% confidence intervals).

[b] The Refined Crouter equation used both 10 sec- and 60 sec-epoch data.

waist-worn AG (2 and 4 km/hr; MAPE (%): 80.0 and 8.3%, respectively) [20]. Although absolute speed measurements during CPET were not utilized in our current study, our findings are partially consistent with the results of Karaca et al. with respect to intensity levels. The ankle-

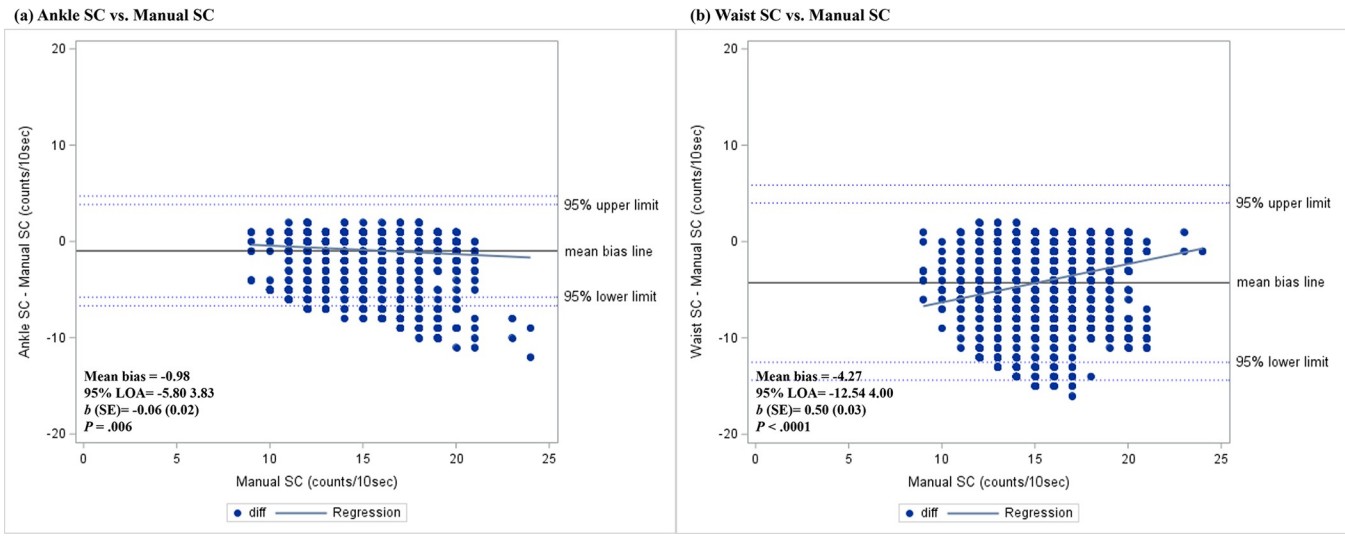

**Fig 3. Bland-Altman plots representing the agreement between ActiGraph-derived step counts and manual step counts.** (a) ankle location; (b) waist location. The horizontal line in the middle represents mean bias, followed by lines representing 95% of the limits of agreement (LOA) and the maximum limits of LOA. The diagonal line represents the linear regression line.

worn AG exhibited agreement with manually counted steps at LPA but underestimated SC at MVPA according to Bland-Altman plots. Conversely, the waist-worn AG significantly underestimated SC at LPA but improved as the intensity level increased. These findings are largely consistent with Bezuidenhout et al., who demonstrated that the ankle-worn AG consistently showed a high level of agreement at overall speeds (0.6–1.4m/s; >94%), while waist-worn AG exhibited poor agreement at lower speeds (0.6–1.0 m/s; 69%) but improved at higher speeds

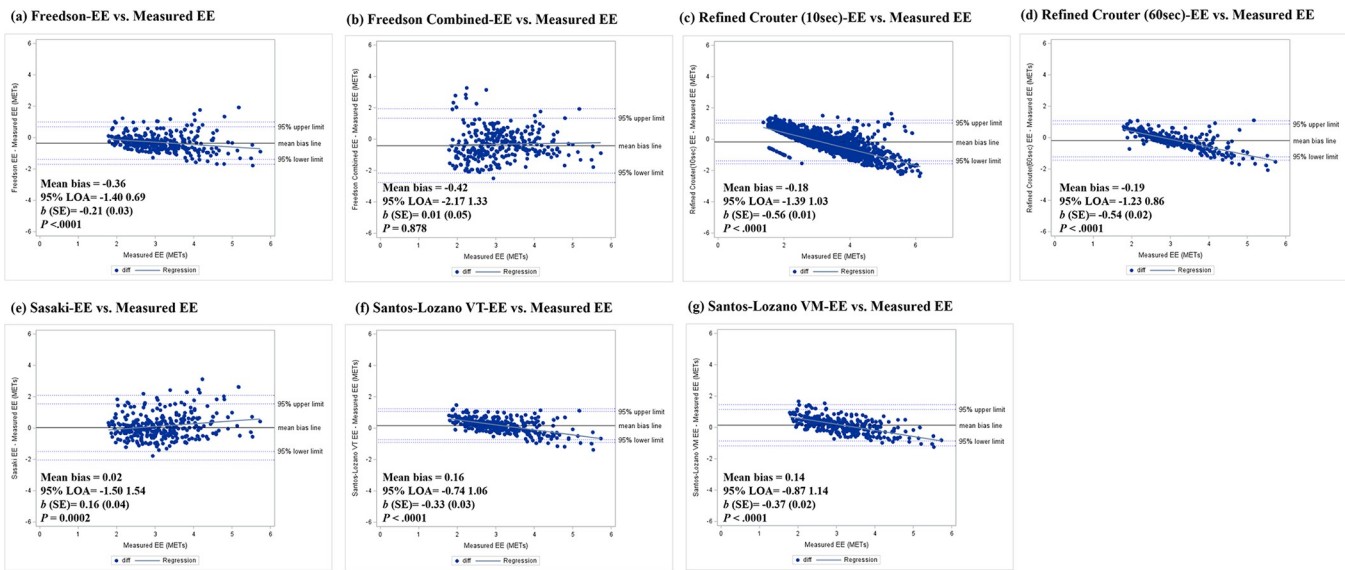

**Fig 4. Bland-Altman plots representing the agreement between ActiGraph-derived and indirect calorimetry-derived energy expenditure using six different equations.** (a) Freedson; (b) Freedson+Williams; (c) Refined Crouter using 10-sec epoch data; (d) Refined Crouter using 60-sec epoch data; (e) Sasaki; (f) Santos-Lozano VT; (g) Santos-Lozano VM. All equations except for the Refined Crouter used 60-sec epoch data. The horizontal line in the middle represents mean bias followed by lines representing 95% of the limits of agreement (LOA) and maximum limits of LOA. The diagonal line represents the linear regression line.

(>1.4 m/s; 94%) [21]. Similar studies examining treadmill exercise tests have also indicated that the waist-worn AG underestimates SC at lower speeds but improves as the speed increases [48, 49]. In the present study, the waist-worn AG had a lower MAPE during MVPA than LPA (LPA: 43.7% vs. MVPA: 16.8%), which is generally consistent with these findings. This is supported by the fact that an AG worn close to the body's center of mass is more sensitive in measuring locomotion resulting from the ambulatory cycle of both legs during active walking or running [17]. However, additional research is necessary to gather more evidence for the discrepancy in the accuracy between LPA and MVPA.

The current study focused on HFpEF patients, a population known to face challenges in performing absolute high-intensity PA due to reduced exercise capacity [50]. HF patients are typically older individuals who experience limitations in walking as a result of aging [51], muscle depletion [52], and frailty, which is independently associated with the risk of HF in older adults [53]. Ozawa et al. found that HF patients experience a significant decrease in walking speed, even after adjusting age and sex-related factors [51]. Therefore, we suggest that ankle-worn AG, which exhibits less variability and bias across different PA intensity levels in SC measurement, would be more appropriate for this population than waist-worn AG.

## Energy expenditure

The current study found that all EE prediction equations showed higher MAPEs overall (>10%) in the HFpEF population. Specifically, the Freedson, refined Crouter, and Santos-Lozano VT/VM equations tended to overestimate EE at LPA but underestimate EE at MVPA when compared to EE derived from indirect calorimetry. However, the Sasaki equation showed the lowest percentage bias overall (0.7%) and at LPA (-1.2%), while the Santos-Lozano VT equation showed the lowest at MVPA (0.2%).

The Freedson equation (1998) was originally developed using the first generation of AG accelerometers (model 7164). Despite the use of an older AG model in the development of this equation, it remains one of the most widely employed equations in PA-related research. However, Grydeland et al. observed that the older AG model (model 7164) tended to overestimate PA outputs compared to the later generation of AG devices [54]. These findings suggest that the Freedson equation may produce less accurate PA estimates when applied to data obtained from the newer AG model. Considering that the Freedson equation showed high MAPE and bias in our study, the later AG model (GT9X) may not be suitable for estimating EE with this equation, at least in adults with HFpEF.

Moreover, studies have reported that the Freedson equation demonstrates accuracy only for PA levels above 1951 counts in young adults and less sensitivity at lower-intensity PA during treadmill tests [22, 23, 55]. Our findings are consistent with these reports, as we observed high MAPE and bias when applying the Freedson equation in the HFpEF population, characterized by lower activity levels and a mean age of over 60 years. The Freedson Combination equation (1998) was developed to overcome the limitation of the Freedson equation by allowing the assessment of lower-intensity activities ($\leq$ 1951 counts) [12, 28]. In the present study, the Freedson Combination equation was the only one that did not exhibit any proportional bias compared to indirect calorimetry, but it showed a wider 95% LOA range compared to other equations. Additionally, this equation still had the highest MAPE overall (28%) and at LPA (40%), with a more significant percentage bias than other equations, indicating large variations and low accuracy for EE estimation. Similarly, Riberio et al. reported that the Freedson Combination equation performed poorly in accurately estimating EE for women with severe obesity requiring clinical treatment [28]. This finding indicates that using this equation to estimate EE may not be advisable for HFpEF patients.

The Crouter equation (2010) was initially developed as a two-regression model in 2006 to differentiate various locomotion activities (i.e., walking/running and lifestyle behaviors) based on the coefficient variations in activity counts and refined in 2010 to improve EE estimation by reducing misclassification of activity types (i.e., walking or running) in free-living settings [24]. In the current study, the refined Crouter equation tended to underestimate EE at MVPA during treadmill testing in the HFpEF population. However, this equation showed varied results in previous studies depending on the study settings and population characteristics. For example, in free-living settings, Crouter et al. reported that it tended to overestimate EE in younger adults [26], but Aguilar-Farias et al. found that it performed more accurately in older adults, particularly at MVPA [14]. In laboratory settings, Rothney et al. found that this equation significantly overestimated total EE compared with a whole-room indirect calorimeter in younger adults [56]. Since the refined Crouter equation was developed based on free-living lifestyle behaviors [24], more evidence of this equation's validity is needed based on the treadmill exercise test. Therefore, further studies are required to validate this equation for clinical populations based on laboratory settings.

Sasaki et al. (2011) developed the equation using AG's VM activity count data to establish cut points and PA intensity classification in young adults. Although few studies have examined the validity of the Sasaki equation, Aguilar-Farias et al. found that this equation showed agreement and better accuracy than other equations when compared with indirect calorimetry in free-living settings among older adults [14]. In our study, the Sasaki equation tended to overestimate EE during MVPA but was the only equation to demonstrate significant equivalence to the criterion EE across all PA intensity levels. The Sasaki equation exhibited better accuracy with the lowest bias overall (mean bias = 0.02; percentage bias = 0.65%) and at LPA (mean bias = -0.07; percentage bias = -1.19%). However, we observed inconsistent outcomes regarding the accuracy of this equation, as the MAPE at LPA was comparatively higher at approximately 20% compared to other equations. MAPE is one of the straightforward methods used to examine accuracy and bias, but it may result in undefined or infinite values when dealing with values close to zero, causing misinterpretation [57, 58]. Compared to MAPE, bias represents the mean error that determines the direction of error, allowing researchers to interpret whether there was an overestimation or underestimation [59]. Moreover, the Bland-Altman plot of the Sasaki equation showed agreement with indirect calorimetry, especially between 2–3 METs, which was comparable with the results of percentage bias at LPA. Therefore, our study concluded that the Sasaki equation generally showed better accuracy, particularly at LPA in patients with HFpEF.

Santos-Lozano et al. (2013) developed two equations to differentiate cut points and classify PA intensities for different age groups, using mean activity counts from three axes (Santos-Lozano VT) and VM outputs from the AG (Santos-Lozano VM) [25]. Similar to other equations, these equations have yielded accurate EE prediction across various PA intensities in young adults in laboratory settings, but their accuracy is limited for older adults [14, 25]. In the current study, we observed that these equations showed the lowest MAPEs (<10%) and biases only at MVPA in HFpEF patients (%bias: Santos-Lozano VT = 0.16%; Santos-Lozano VM = -1.34%). Indeed, based on the Bland-Altman plots, the Santos-Lozano equations demonstrated agreement with the criterion measure around the 3 METs level, but they significantly underestimated EE as the intensity increased. This result may have been influenced by the current study's categorization of PA intensity, which combined moderate and vigorous activity into one category of MVPA. Given that our study population consisted of clinical HFpEF patients, who are often limited in their ability to perform activity at the typical absolute vigorous-intensity threshold, we suggest that future studies consider classifying PA intensity levels differently, taking into account the characteristics typically observed in this clinical population.

## Strengths and limitations

To the best of our knowledge, this is the first study to examine the criterion validity of the AG accelerometer among patients with HFpEF. Thus, our study may have meaningful implications for the efficient use of AG accelerometers in this clinical population. Additionally, our study used the later generation of the AG, the GT9X model, which is equipped with triaxial sensors and improved algorithms capable of providing more comprehensive and accurate activity measurements than the earlier models (e.g., Model GT1M and 7164), which was limited to a uniaxial feature [12]. Moreover, our study was conducted in a well-controlled laboratory setting using treadmill exercise testing and compared the accuracy of AG-derived measurements to the gold standard criterion-derived measurements across different PA intensity levels. This controlled environment allowed for a more standardized assessment of the validity of the AG-GT9X in estimating PA parameters.

Nevertheless, there are several limitations to be acknowledged in this study. First, the majority of participants were female, with only one male included, which limits the generalizability of our findings to the entire HFpEF population, particularly for male HF patients. Future studies with large sample sizes and more equal distribution of both genders are required to enhance the generalizability of the results in this clinical population. Another limitation is the reliance on standard resting EE (3.5 mL/kg/min) instead of measured resting EE for the calculation of METs values, which is considered a more accurate indicator of resting metabolic rate for the calculation of METs values [60]. A recent study found that the discrepancy between measured and estimated resting EE becomes more pronounced as BMI and age increase among individuals with HFpEF [61]. Considering our study population consists of adults with HFpEF whose mean BMI is greater than 38 and whose mean age is above 60 years, it is advisable to measure resting EE before the test to obtain more reliable results. Therefore, future studies should incorporate the measured resting EE to ensure the validity of the findings. In addition, the present study categorized PA levels using the general PA intensity classification (i.e., LPA: 1.5–2.99 METs; MVPA $\geq$ 3 METs) [39], which may not accurately reflect the PA intensity relative to the low fitness levels of the HFpEF population. Future studies should consider utilizing relative PA intensity cut-points developed specifically for HFpEF patients to improve the accuracy of PA level classification [62]. Among several different EE estimation equations, we used an adapted version of the refined Crouter equation, particularly developed for walking/running activities, using both 10-sec and 60-sec epoch data; yet this modified cut-point scaling may not adequately account for the magnitude or coefficient variation of the counts during the treadmill test [63]. Further research is warranted to explore the use of coefficient variation in determining the most accurate refined Crouter equation for activities rather than relying on a single equation. Moreover, since this study was designed based on a laboratory setting using treadmill exercise testing, the validity of the AG in free-living conditions may differ. Given that a few EE estimation equations used in this study were originally developed for different types of activities (i.e., free-living activities), future research should incorporate a range of activities in addition to CPET and aim to explore the validity of the AG in real-life free-living conditions. Lastly, we could not consider the impact of absolute walking speed on the accuracy of the AG during exercise testing in this study due to the colinear increases in treadmill grade. Instead, we calculated the PA intensity level using EE estimation obtained during CPET and assessed the accuracy of AG at the corresponding PA intensity levels. Still, we must acknowledge that this approach is not specifically tailored for CPET data. Further investigations should explore the impact of absolute walking speed on the accuracy of the AG to better understand its validity across different exercise intensities during CPET, while considering real-world variability.

## Conclusion

In conclusion, our study found that the accuracy of measuring SC was superior with the ankle-worn AG than the waist-worn AG in laboratory conditions among adults with HFpEF. Despite a majority of AG-EE prediction equations displaying poor accuracy in this HFpEF population, the Sasaki equation demonstrated better accuracy overall and particularly at LPA. However, further research is still warranted to cross-validate the results in different environments. Nonetheless, these findings offer valuable insight for future clinical research when selecting the most appropriate AG placement and EE prediction equations tailored to specific study populations and study settings. Furthermore, this study may provide crucial guidance for utilizing ankle-worn AG to assess SC among HFpEF patients in clinical research settings where established recommendations for AG placement are lacking. This will enhance the validity and reliability of PA assessment, contributing to the overall quality of future studies in this population.

## Supporting information

**S1 Table. Glossary of abbreviations.**
(DOCX)

**S2 Table. Bland-Altman analysis for step counts and energy expenditure in each location and algorithm.**
(DOCX)

**S3 Table. Total step counts and energy expenditure during cardiopulmonary treadmill test using raw data.**
(DOCX)

**S4 Table. Accuracy and agreement of AG-derived step counts and energy expenditures compared to criterion measures using raw data.**
(DOCX)

**S1 File.**
(XLSX)

## Acknowledgments

All authors would like to thank all patients who participated in the present study.

## Author Contributions

**Data curation:** Jisu Kim, Jonathan Kenyon, Hayley Billingsley, Natalie Bohmke, Syed Imran Ahmed, Hannah Salmons, Danielle Kirkman, Salvatore Carbone, Youngdeok Kim.

**Formal analysis:** Jisu Kim, Jonathan Kenyon, Youngdeok Kim.

**Investigation:** Jisu Kim, Jung-Min Lee, Youngdeok Kim.

**Methodology:** Jisu Kim, Jung-Min Lee, Danielle Kirkman, Salvatore Carbone, Youngdeok Kim.

**Project administration:** Youngdeok Kim.

**Writing – original draft:** Jisu Kim.

**Writing – review & editing:** Jonathan Kenyon, Hayley Billingsley, Natalie Bohmke, Syed Imran Ahmed, Hannah Salmons, Jung-Min Lee, Danielle Kirkman, Salvatore Carbone, Youngdeok Kim.

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
