## [Decision Letter · Decision Letter 0]

30 Jul 2024

PONE-D-24-20379Validity of the ActiGraph-GT9X accelerometer for measuring steps and energy expenditures in heart failure patientsPLOS ONE

Dear Dr. Kim,

Thank you for submitting your manuscript to PLOS ONE. After careful consideration, we feel that it has merit but does not fully meet PLOS ONE’s publication criteria as it currently stands. Therefore, we invite you to submit a revised version of the manuscript that addresses the points raised during the review process.

A research study was conducted to examine the criterion validity of the AG-GT9X for measuring step counts (SC) and energy expenditure (EE) among heart failure patients, with repeated measures taken from 16 participants. Reviewer #1 pointed out the inappropriate use of statistical analysis for repeated-measures data, suggesting a comprehensive re-analysis and appropriate methods for summarizing data. Reviewer #2 appreciated the manuscript's validation of the ActiGraph accelerometer in laboratory settings but noted the need for further clinical validation. Reviewer #3 provided detailed comments on various sections, suggesting clarifications and revisions to improve the manuscript's clarity and accuracy, including specifying the sample size, device details, and statistical methods. The reviewers (especially Reviewer #3) collectively highlighted the need for improved data analysis, accurate device descriptions, and clear methodological explanations to enhance the study's validity and applicability. 

We look forward to receiving your revised manuscript.

Kind regards,

Jyotindra Narayan

Academic Editor

PLOS ONE

Journal Requirements:

Reviewers' comments:

Reviewer's Responses to Questions

**Comments to the Author**

1. Is the manuscript technically sound, and do the data support the conclusions?

Reviewer #1: Yes

Reviewer #2: Yes

Reviewer #3: Partly

2. Has the statistical analysis been performed appropriately and rigorously? 

Reviewer #1: No

Reviewer #2: Yes

Reviewer #3: Yes

3. Have the authors made all data underlying the findings in their manuscript fully available?

Reviewer #1: No

Reviewer #2: Yes

Reviewer #3: No

4. Is the manuscript presented in an intelligible fashion and written in standard English?

Reviewer #1: Yes

Reviewer #2: Yes

Reviewer #3: Yes

5. Review Comments to the Author

Reviewer #1: A research study was conducted which examined the criterion validity of the AG-GT9X for measuring step counts (SC) and energy expenditure (EE) among heart failure patients. Data from some patients was repeatedly measured: 41 CPET from 16 participants. The results are unclear because the statistical analysis is inappropriate for analyzing repeated-measures data.

Major revisions:

Overall, a comprehensive re-analysis of the repeated measures data is needed.

1- Since data collection occurred repeatedly for some participants the analysis should be tailored for repeated measures data. Thus Pearson’s correlation and t-tests are not appropriate in a repeated-measure study as it ignores the correlation of the outcomes from multiple visits within the same participant. Furthermore, the calculation of standard deviation deviates from the conventional method when repeated measures are performed.

2- The standard statistical term for average is mean.

3- Table 1: For continuous data, if it is normally distributed, summarize as means and standard deviations. If the distribution of the data violates the normality assumption, summarize as medians, first and third quartiles.

Reviewer #2: Thank you for submitting this work.

The manuscript is validating the use of the ActiGraph accelerometer to assess physical activity in heart failure patients. They have examined step counts and energy expenditure using this accelerometer. They found out that this accelerometer successfully showed better accuracy in laboratory settings but need to be validated in other clinical settings.

Body of manuscript is written good. No further comments to revise it.

Reviewer #3: General Comments:

PONE-D-24-20379 presents a validation of device-based energy expenditure estimation methods and step counts in a sample of patients with heart failure. There are several areas that need attention. Please see specific comments and suggestions by section of the manuscript below.

The references have some inconsistencies. Please complete a thorough check.

Abstract:

Is the Total # here the number of patients or the number of tests? Please consider clarifying. I encourage the authors to present the N of patients (mean age +/- sd) and then perhaps the median (IQR) number of tests. I find that more relevant than the total number of tests.

Which ankle was the GT9X worn on? Please consider specifying if adequate space is available

Introduction:

• Lines 17-18: please consider revising this to reflect that ActiGraph is a device manufacturer and that there are multiple device models available. “The Actigraph” reads to me as if it is a singular device type which may be misleading for some readers.

• Lines 18-22: please consider revising for clarity. The devices collect high temporal resolution triaxial acceleration that is commonly summarized by calculating a vector magnitude. This is not limited to just habitual locomotor activities. Similar to my comment above, I encourage the authors to be explicit for readers that may not be as familiar with the device-based literature. It may also be worthwhile to mention how counts factor in to this as that may not be understood by all.

• Lines 22-26: similar to my above comments, I encourage the authors to give more context about these summary measures derived from the acceleration data. Steps and EE are just two of many device-based PA outcomes to be obtained from these data. I think this paragraph could benefit from additional detail.

• Lines 27-30: this sentence may be enhanced by adding something about the outcome of interest also being a factor. This may make it more clear to the reader why the ankle is generally better for steps and the waist for EE, etc.

• Lines 31-34: Ref 16 duplicated. Please check.

• Lines 41-47: I would consider adding in here that these and many of the EE equations were specifically developed using data from waist-worn devices.

• Lines 64-66: should this be “indirect calorimetry measured EE”?

Methods:

• Lines 80-84: With an N of 16 and 15/16 coming from one of two clinical trials where obesity was also an inclusion factor raises the question of why two clinical trials and why these data were pooled? That isn’t clear at present.

• Lines 96-97: this difference further brings up question as to why the data from the two trials were pooled, especially with the addition of only one additional participant.

• Lines 97-99: please consider moving this to the results.

• Lines 102-104: please consider moving this after Lines 104-106 or integrate the two. This reads as if the protocol was ramped on METs but that is the response the actual ramp (speed and grade) was intended to achieve.

• Lines 109-111: If you eliminate the last 5% of the data, might that include the VO2peak data? This may be dependent on the interval these data are output in so please consider specifying that aspect. The two citations provided here are for steady state data which is considerably different from CPET data where a plateau isn’t likely so I’m not sure what the authors mean by variability below 5% for a plateau here.

• Lines 122-125: The GT9X is not the latest generation of ActiGraph accelerometers and is in fact no longer in production by the manufacturer (https://blog.theactigraph.com/blog/gt9x-link) and there are several newer device generations currently available (https://theactigraph.com/wearable-devices).

• Line 124: the vertical, anteroposterior, and mediolateral orientations are generally used for devices worn on the waist, and while they also generally apply to the ankle, they should probably be listed as X, Y (vertical on waist), and Z.

• Lines 125-126: was the optional Idle Sleep Mode feature on or off for these data collections? It probably won’t have much bearing on these data since the authors use counts and not raw data but please consider specifying for transparency.

• Lines 126-128: Please consider revising for clarity. The raw .gt3x files were downloaded and converted to counts in the specified epochs. Were these counts data converted from the raw using the normal filter or low frequency extension (LFE)? This is important for steps and counts so please specify.

o Cain KL, Conway TL, Adams MA, Husak LE, Sallis JF. Comparison of older and newer generations of ActiGraph accelerometers with the normal filter and the low frequency extension. Int J Behav Nutr Phys Act. 2013 Apr 25;10:51. doi: 10.1186/1479-5868-10-51.

o Feito Y, Hornbuckle LM, Reid LA, Crouter SE. Effect of ActiGraph's low frequency extension for estimating steps and physical activity intensity. PLoS One. 2017 Nov 20;12(11):e0188242. doi: 10.1371/journal.pone.0188242

o Toth, L. P., Park, S., Pittman, W. L., Sarisaltik, D., Hibbing, P. R., Morton, A. L., ... & Bassett, D. R. (2018). Validity of activity tracker step counts during walking, running, and activities of daily living. Translational Journal of the American College of Sports Medicine, 3(7), 52-59.

o Toth LP, Park S, Springer CM, Feyerabend MD, Steeves JA, Bassett DR. Video-Recorded Validation of Wearable Step Counters under Free-living Conditions. Med Sci Sports Exerc. 2018 Jun;50(6):1315-1322. doi: 10.1249/MSS.0000000000001569.

o Toth, L. P., Park, S., Pittman, W. L., Sarisaltik, D., Hibbing, P. R., Morton, A. L., ... & Bassett, D. R. (2019). Effects of brief intermittent walking bouts on step count accuracy of wearable devices. Journal for the Measurement of Physical Behaviour, 2(1), 13-21.

• Lines 142-144: interesting. I’m not sure I’ve ever seen the Crouter two regression method split like this but I suppose it makes some sense in this case. I’d just make it clear that this isn’t generally the implementation of this method. I suggest the authors call it an adapted version of the method for clarity. The CV is used to determine which of the two equations are used and in this case you’re forcing this to be only 1 equation regardless of the counts magnitude or CV. To be fair, with only treadmill walking and running, that would be the likely outcome but this is brute forced into that equation and not determined as the developer intended.

• Lines 149-151: I understand the need to do this in this study, but cut-point scaling is generally not recommended. This may warrant mention in the discussion/limitations so that is apparent to readers.

o Hibbing PR, Bassett DR, Crouter SE. Modifying Accelerometer Cut-Points Affects Criterion Validity in Simulated Free-Living for Adolescents and Adults. Res Q Exerc Sport. 2020 Sep;91(3):514-524. doi: 10.1080/02701367.2019.1688227.

• Table 1: I’m not sure I understand the “b” superscript here. What does the Crouter two regression method have to do with steps counts? This method is for EE not step counting.

Results:

• Lines 207-208: out of curiosity, was the one male from the second clinical trial? If so, that further raises the point about pooling the data from the two trials for this analysis. The authors can’t generalize these findings as it is a nearly all female study and may be better presented that way.

• Lines 208-209: please define NYHA in the text.

• Lines 230-234: I’m not sure how much value the correlation analysis adds. I think it would be generally expected these measures would correlate with the criterion measures.

• Table 3: I’m not sure I understand how the intensity-specific step counts were achieved. At what temporal resolution were these data aligned? There isn’t a clean temporal indication of the switch from LPA to MVPA as this may have happened mid-interval so this couldn’t be cleanly aligned at the 30s or 1-minute level, correct?

Discussion:

• Lines 335-336: please see also. This is why it is important to state whether your step counts come from the normal filter or LFE as that directly affects interpretation of these results.

o Toth, L. P., Park, S., Pittman, W. L., Sarisaltik, D., Hibbing, P. R., Morton, A. L., ... & Bassett, D. R. (2019). Effects of brief intermittent walking bouts on step count accuracy of wearable devices. Journal for the Measurement of Physical Behaviour, 2(1), 13-21.

• Lines 376-380: please change language about GT9X being the latest/newest/most recent ActiGraph device model.

• Lines 447-450: please change language about GT9X being the latest/newest/most recent ActiGraph device model. Additionally, the GT3X devices were also triaxial devices. The prior GT1M and 7164 models were all uniaxial. Please revise.

• Lines 470-471: The authors may have the opportunity in future work with these data to explore relative intensity with these patients since traditional absolute intensity measures may not be particularly helpful in the clinic or free-living evaluation of their physical behaviours. This may overcome some of the interpretation limitations the authors mention.

o Rowlands, A. V., Orme, M. W., Maylor, B., Kingsnorth, A., Herring, L., Khunti, K., ... & Yates, T. (2023). Can quantifying the relative intensity of a person’s free-living physical activity predict how they respond to a physical activity intervention? Findings from the PACES RCT. British Journal of Sports Medicine, 57(22), 1428-1434.

o Orme, M. W., Lloyd-Evans, P. H., Jayamaha, A. R., Katagira, W., Kirenga, B., Pina, I., ... & Rowlands, A. V. (2023). A case for unifying accelerometry-derived movement behaviors and tests of exercise capacity for the assessment of relative physical activity intensity. Journal of Physical Activity and Health, 20(4), 303-310.

o Kingsnorth, A. P., Rowlands, A. V., Maylor, B. D., Sherar, L. B., Steiner, M. C., Morgan, M. D., ... & Orme, M. W. (2022). A more intense examination of the intensity of physical activity in people living with chronic obstructive pulmonary disease: insights from threshold-free markers of activity intensity. International Journal of Environmental Research and Public Health, 19(19), 12355.

• Lines 473-474: the authors should consider mentioning as a limitation that the application in which these EE methods were applied is also not generally the intended use case having data from only a CPET instead of additional activities like those these methods were developed with.

6. PLOS authors have the option to publish the peer review history of their article (what does this mean?). If published, this will include your full peer review and any attached files.

Reviewer #1: No

Reviewer #2: **Yes: **Alka Bishnoi

Reviewer #3: No

---

## [Decision Letter · Decision Letter 1]

4 Oct 2024

PONE-D-24-20379R1Validity of the ActiGraph-GT9X accelerometer for measuring steps and energy expenditures in heart failure patientsPLOS ONE

Dear Dr. Kim,

Thank you for submitting your manuscript to PLOS ONE. After careful consideration, we feel that it has merit but does not fully meet PLOS ONE’s publication criteria as it currently stands. Therefore, we invite you to submit a revised version of the manuscript that addresses the points raised during the review process.

The manuscript assesses the ActiGraph-GT9X's accuracy in tracking step count and energy expenditure in HFpEF patients. Reviewer #3 raised concerns about data exclusion possibly omitting VO₂peak values, given the lack of a steady state in ramped CPET tests. Reviewer #4 highlighted areas for improvement, such as the sample’s gender imbalance, overestimation of EE at low activity levels, and unaddressed speed variations affecting accuracy. They also recommended expanding on the interpretation of MAPE, bias, proportional bias from Bland-Altman plots, and limitations of using standard resting EE for HFpEF patients, urging greater methodological clarity and clinical relevance. Finally, I would recommend authors to provide the data used in public domain to improve the transparency and repeatibility of the work.

We look forward to receiving your revised manuscript.

Kind regards,

Jyotindra Narayan

Academic Editor

PLOS ONE

Reviewers' comments:

Reviewer's Responses to Questions

**Comments to the Author**

1. If the authors have adequately addressed your comments raised in a previous round of review and you feel that this manuscript is now acceptable for publication, you may indicate that here to bypass the “Comments to the Author” section, enter your conflict of interest statement in the “Confidential to Editor” section, and submit your "Accept" recommendation.

Reviewer #1: All comments have been addressed

Reviewer #3: (No Response)

Reviewer #4: All comments have been addressed

2. Is the manuscript technically sound, and do the data support the conclusions?

Reviewer #1: (No Response)

Reviewer #3: Partly

Reviewer #4: Partly

3. Has the statistical analysis been performed appropriately and rigorously? 

Reviewer #1: (No Response)

Reviewer #3: Yes

Reviewer #4: Yes

4. Have the authors made all data underlying the findings in their manuscript fully available?

Reviewer #1: (No Response)

Reviewer #3: No

Reviewer #4: Yes

5. Is the manuscript presented in an intelligible fashion and written in standard English?

Reviewer #1: (No Response)

Reviewer #3: Yes

Reviewer #4: Yes

6. Review Comments to the Author

Reviewer #1: (No Response)

Reviewer #3: Methods:

• Lines 168-170: If you eliminate the last 5% of the data, might that include the VO2peak data? This may be dependent on the interval these data are output in so please consider specifying that aspect. The two citations provided here are for steady state data which is considerably different from CPET data where a plateau isn’t likely so I’m not sure what the authors mean by variability below 5% for a plateau here.

o Author Response: We manually reviewed the initial raw data to determine an optimal range for data exclusion to minimize variability at the beginning and end of the test and retain a steady state during the CPET. We considered potential sources of variability that could not be controlled, such as participants holding onto the treadmill or accidentally stopping before clock time stopped which could disrupt accurate measure. We also confirmed that eliminating 5% of the data, which was approximately 30 seconds at both the start and end of the test, did not significantly affect individuals' VO₂peak measurements. To clarify, we revised the sentence. – lines 168-170: “After thoroughly reviewing each participant’s raw dataset, the first and last 5% of the data were eliminated before analysis and retained a steady state during CPET to minimize the variability from the potential uncontrollable sources [14, 36]”

o Reviewer Comment: This remains unclear. With a ramped CPET to achieve VO2peak, steady state is not possible. This is why the test terminates at volitional fatigue because the participant cannot meet the demand.

Reviewer #4: The manuscript explores the validity of the ActiGraph-GT9X accelerometer for measuring step counts and energy expenditure in heart failure patients. The research topic is important given the clinical implications of accurate physical activity measurement in managing heart failure with preserved ejection fraction (HFpEF). While the study provides meaningful contributions, there are several areas where clarity, methodological rigor, and interpretation of results can be improved.

1. The sample’s gender imbalance (15 females, 1 male) limits generalizability. Discuss its impact and recommend future studies with more balanced gender representation.

2. The Freedson and Sasaki equations overestimate EE at low PA levels. Further clarify how HFpEF characteristics may influence these discrepancies .

3. Speed variations were not accounted for, which may affect step count accuracy. Add discussion about the role of walking speeds, and recommend future studies involving real-world variability.

4. Provide more interpretation of MAPE and bias, particularly their clinical relevance for HFpEF patients. Discuss acceptable bias levels and their implications for step count and EE measurements.

5. Bland-Altman plots reveal proportional bias. Expand the discussion of the limits of agreement and clarify how proportional bias affects the accuracy of AG-derived measures.

6. Using a standard resting EE (3.5 ml/kg/min) may not be appropriate for HFpEF patients. Emphasize this limitation and suggest using measured resting EE in future studies.

7. PLOS authors have the option to publish the peer review history of their article (what does this mean?). If published, this will include your full peer review and any attached files.

Reviewer #1: No

Reviewer #3: No

Reviewer #4: **Yes: **Subhash Pratap

---

## [Decision Letter · Decision Letter 2]

21 Oct 2024

PONE-D-24-20379R2Validity of the ActiGraph-GT9X accelerometer for measuring steps and energy expenditures in heart failure patientsPLOS ONE

Dear Dr. Kim,

Thank you for submitting your manuscript to PLOS ONE. After careful consideration, we feel that it has merit but does not fully meet PLOS ONE’s publication criteria as it currently stands. Therefore, we invite you to submit a revised version of the manuscript that addresses the points raised during the review process.

**After carefully reading the manuscript and reviewer comments, it is observed that the authors fail to address the concerns raised by the reviewers, especially, in presenting the dataset in public domain for better repeatability. There remains an issue with data availability in regards to journal standards. The author’s state the data are available in the SIF but they are not. The SIF is simply tables of data presented in the manuscript. Therefore, the manuscript cannot be accepted in its current form. The authors are encouraged to revise the manuscript in accordance with the previous reviewer comments and make the complete raw and processed data publicly accessible, rather than providing it solely as a supplementary table, to enhance transparency and reproducibility. **

We look forward to receiving your revised manuscript.

Kind regards,

Jyotindra Narayan

Academic Editor

PLOS ONE

Reviewers' comments:

Reviewer's Responses to Questions

**Comments to the Author**

1. If the authors have adequately addressed your comments raised in a previous round of review and you feel that this manuscript is now acceptable for publication, you may indicate that here to bypass the “Comments to the Author” section, enter your conflict of interest statement in the “Confidential to Editor” section, and submit your "Accept" recommendation.

Reviewer #1: All comments have been addressed

Reviewer #3: All comments have been addressed

Reviewer #4: (No Response)

2. Is the manuscript technically sound, and do the data support the conclusions?

Reviewer #1: (No Response)

Reviewer #3: Yes

Reviewer #4: (No Response)

3. Has the statistical analysis been performed appropriately and rigorously? 

Reviewer #1: (No Response)

Reviewer #3: Yes

Reviewer #4: (No Response)

4. Have the authors made all data underlying the findings in their manuscript fully available?

Reviewer #1: (No Response)

Reviewer #3: No

Reviewer #4: (No Response)

5. Is the manuscript presented in an intelligible fashion and written in standard English?

Reviewer #1: (No Response)

Reviewer #3: Yes

Reviewer #4: (No Response)

6. Review Comments to the Author

**Reviewer #1: **(No Response)

**Reviewer #3:** (No Response)

**Reviewer #4:** The revised manuscript does not sufficiently address these major concerns, and therefore it is not recommended for acceptance.

7. PLOS authors have the option to publish the peer review history of their article (what does this mean?). If published, this will include your full peer review and any attached files.

Reviewer #1: No

Reviewer #3: No

Reviewer #4: No

---

## [Author Response · Author response to Decision Letter 2]

23 Oct 2024

In response to the academic editor's comment, we have ensured that the de-identified raw data are fully accessible without restrictions in the Supporting Information File and have updated the data availability statement in the manuscript accordingly. Additionally, we have retained the response letter from the previous revision to confirm that all reviewer comments have been addressed. Finally, we have included an overall reflection on the efforts made to address the reviewers' feedback throughout the entire revision process.

---

## [Decision Letter · Decision Letter 3]

28 Nov 2024

Validity of the ActiGraph-GT9X accelerometer for measuring steps and energy expenditures in heart failure patients

PONE-D-24-20379R3

Dear Dr. Kim,

We’re pleased to inform you that your manuscript has been judged scientifically suitable for publication and will be formally accepted for publication once it meets all outstanding technical requirements.

Kind regards,

Jyotindra Narayan

Academic Editor

PLOS ONE

Additional Editor Comments (optional):

The reviewers have seen the revised manuscript and now recommended for publication. Congratulations to the authors for submitting the quality work.

Reviewers' comments:

Reviewer's Responses to Questions

**Comments to the Author**

1. If the authors have adequately addressed your comments raised in a previous round of review and you feel that this manuscript is now acceptable for publication, you may indicate that here to bypass the “Comments to the Author” section, enter your conflict of interest statement in the “Confidential to Editor” section, and submit your "Accept" recommendation.

Reviewer #3: All comments have been addressed

Reviewer #4: All comments have been addressed

2. Is the manuscript technically sound, and do the data support the conclusions?

Reviewer #3: Yes

Reviewer #4: Yes

3. Has the statistical analysis been performed appropriately and rigorously? 

Reviewer #3: Yes

Reviewer #4: Yes

4. Have the authors made all data underlying the findings in their manuscript fully available?

Reviewer #3: Yes

Reviewer #4: Yes

5. Is the manuscript presented in an intelligible fashion and written in standard English?

Reviewer #3: Yes

Reviewer #4: Yes

6. Review Comments to the Author

Reviewer #3: (No Response)

Reviewer #4: The authors have addressed all the concerns raised by the reviewer. Happy to see they have also provided the raw data. The manuscript can be accepted without any further revisions.

7. PLOS authors have the option to publish the peer review history of their article (what does this mean?). If published, this will include your full peer review and any attached files.

Reviewer #3: No

Reviewer #4: No

---

## [Editor Report · Acceptance letter]

8 Dec 2024

PONE-D-24-20379R3 

PLOS ONE

Dear Dr. Kim, 

I'm pleased to inform you that your manuscript has been deemed suitable for publication in PLOS ONE. Congratulations! Your manuscript is now being handed over to our production team.

Kind regards, 

on behalf of

Dr. Jyotindra Narayan 

Academic Editor

PLOS ONE